# Investigating Instruction Tuning Large Language Models on Graphs

**Kerui Zhu\*, Bo-Wei Huang\*, Bowen Jin\***[*]**, Yizhu Jiao, Ming Zhong, Kevin Chang**
**Shou-De Lin, Jiawei Han**
University of Illinois at Urbana-Champaign, National Taiwan University
{keruiz2, boweiwh2, bowenj4}@illinois.edu

## Abstract

Inspired by the recent advancements of Large Language Models (LLMs) in NLP tasks, there's growing interest in applying LLMs to graph-related tasks. This study delves into the capabilities of instruction-following LLMs for engaging with real-world graphs, aiming to offer empirical insights into how LLMs can effectively interact with graphs and generalize across graph tasks. We begin by constructing a dataset designed for instruction tuning, which comprises a diverse collection of 79 graph-related tasks from academic and e-commerce domains, featuring 44,240 training instances and 18,960 test samples. Utilizing this benchmark, our initial investigation focuses on identifying the optimal graph representation that serves as a conduit for LLMs to understand complex graph structures. Our findings indicate that JSON format for graph representation consistently outperforms natural language and code formats across various LLMs and graph types. Furthermore, we examine the key factors that influence the generalization abilities of instruction-tuned LLMs by evaluating their performance on both in-domain and out-of-domain graph tasks.[1]

## 1 Introduction

The success of Large language models (LLMs) in understanding and reasoning the semantic structure in natural language has brought a great interest in applying this capability to assist tasks with other modalities such as graphs. Graph stores information through the explicit connections between nodes and the attributes associated with the nodes and edges, which is quite different from natural language. To fill the gap between graph and LLM, Ye et al. (2023); Wang et al. (2024b); He & Hooi (2024); Luo et al. (2024) focus on instruction tuning LLM on linearized graph representations so that the LLM can learn the graph structure and solve graph-related tasks based on instructions. Results show that graph instruction-tuned LLMs outperform traditional Graph Neural Networks (GNNs) (Ye et al., 2023).

However, we notice a lack of fundamental study of the graph representation and deeper analysis of the instruction-tuned LLM's generalization ability over empirical graph tasks. For the graph representation, Chen et al. (2023); Zhao et al. (2023); Wang et al. (2024a) translate graphs into natural language, while Wang et al. (2024b) represents the graph in a code-like format. However, it is still unclear how the choice of graph representation would affect the efficiency of graph instruction tuning. For the generalization, the LLMs are expected to solve tasks with new requirements, new graph structure distribution, and even unseen algorithms. This is critical for a general graph problem solver due to the complexity and variety of graph-related problems. Guo et al. (2023) establish a benchmark with 10 tasks to assess the proficiency of LLMs in understanding graph data. In this work, we instruction-tune LLMs on graphs with more fine-grained tasks and comprehensively analyze their capability concerning generalization.

---

[*]The first three authors contribute equally.
[1]Code is available at https://github.com/ZhuKerui/graph-instruction-tuning

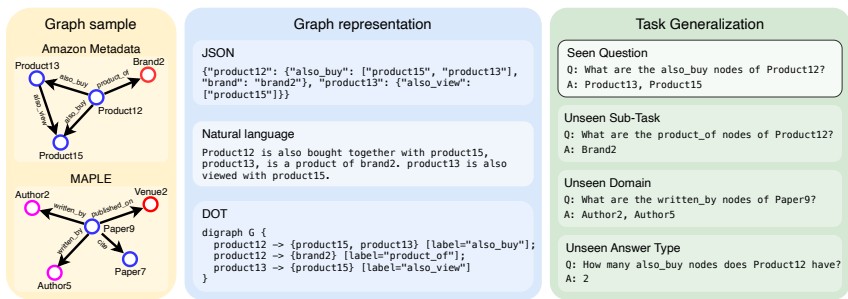

Figure 1: Examples of graph representations and three levels of generalization.

To facilitate the analysis of generalization, we build a benchmark for graph instruction tuning consisting of 14 tasks with 7 categories. We further derive 79 sub-tasks from the 14 tasks and sample 63.2k question-graph pairs from academic and e-commerce networks.

In general, our work focuses on two research questions regarding graph instruction-tuning:

- **RQ1: What is the optimal graph representation that serves as a conduit for LLMs to learn graph structures effectively?** For the first question, we experiment with three types of graph representation: natural language, structured text representation (JSON), and code (DOT). Figure 1 shows an example of each representation. We instruction-tune LLMs with each representation separately and evaluate their performance on our constructed benchmark. The result shows the JSON format offers the best performance across various LLMs and graph domains.

- **RQ2: To what extent can LLMs fine-tuned on limited graph-related tasks generalize to unseen tasks?** To further analyze the generalization of the graph instruction-tuned model, we propose three levels of generalization, namely *Unseen Sub-Task*, *Unseen Domain*, and *Unseen Answer Type*, where each level tests the LLM on a scenario not seen during the training. Figure 1 shows an example question at each level. We conduct comprehensive experiments on the LLM's capability to the three levels of generalization. The result indicates that the LLM can generally be improved over a wide range of graph-related tasks after a limited graph instruction tuning. However, the LLM may easily get overfitted on simple counting tasks and doesn't generalize well on inductive reasoning tasks like link prediction. In addition, our findings also reveal LLM is capable of handling tasks requiring graph algorithms not seen during training, which indicates that graph instruction tuning enables LLM to derive new algorithms itself based on its understanding of the graph and the algorithms learned during training.

To summarize, our key contributions are threefold:

- We create a benchmark with fine-grained tasks from two different domain networks, which allows a comprehensive study of the generalization problem.

- We investigate the influence of different graph representations in graph instruction tuning, including natural language, JSON, and code. The result shows the JSON format gives the LLMs the best performance after tuning.

- We propose three levels of generalization for graph-related tasks and conduct extensive experiments to investigate the generalization of the instruction-tuned LLMs. Our experiments reveal tasks where LLM could be overfitted or hard to generalize. Our experiments also show LLM can derive algorithms from the learned algorithms.

## 2 Related Work

### 2.1 LLMs on Graphs

Inspired by the recent achievements of large language models (LLMs) in natural language processing tasks, researchers are investigating the use of LLMs for tackling graph-related tasks (Jin et al., 2023a). Existing works can be organized into two categories depending on the functions of LLMs. The first category typically relies on LLMs to serve as pretrained feature extractors (Chien et al., 2021) for graph neural networks (GNNs) (Wu et al., 2020) . For example, TextGNN (Zhu et al., 2021) proposes to conduct LLM text feature extraction before GNNs for sponsored search tasks. TAPE (He et al., 2023) adopts LLMs to generate augmented texts before feeding into medium-scale LMs and GNNs. The second category is graph-incorporated LLM architectures (Jin et al., 2023b). Specifically, GraphFormers (Yang et al., 2021) and Edgeformers (Jin et al., 2023c) propose graph-empowered language model architecture for homogeneous text-attributed graphs and textual-edge graphs respectively. Heterformer (Jin et al., 2023d) further introduces textless node encoding and proposes an architecture for heterogeneous text-attributed graphs. However, most existing works mainly focus on applying LLMs as off-the-shelf encoders or exploring LLM architecture improvement for graphs. In our work, we investigate the problem of instruction tuning large language models on graphs.

### 2.2 Instruction Tuning for LLMs

Instruction tuning (Ouyang et al., 2022; Sanh et al., 2022) is crucial for the latest generation of LLMs to cater to explicit user commands. In this stage, LLMs are trained using datasets with specific instructions and the expected responses, which improves LLMs in understanding and reacting to various human queries in natural language. Instruction tuning can be seen as a form of meta-learning where the model learns to adapt using the instructions (Zhang et al., 2023a; Longpre et al., 2023). As a result, these models acquire zero-shot learning ability which emerges as natural interactions with users. Currently, this paradigm has already demonstrated its impressive effectiveness across a wide range of natural language tasks, such as coding generation (Luo et al., 2023), complex reasoning (Mukherjee et al., 2023), information extraction (Jiao et al., 2023), and creative writing (Li et al., 2023).

Inspired by the success of instruction tuning on texts, recently an increasing research interest has tried to enable LLMs to generate more accurate and contextually appropriate responses for graph-structured data. These works typically align the language capacity of LLMs with the nuances of graph learning tasks. Specifically, Ye et al. (2023) instruction tunes the LLMs to perform graph tasks with graph structure described in natural language through highly scalable prompts. Wang et al. (2024b) uses a code-like format to describe graph information. Luo et al. (2024) is a concurrent work closest to ours. It instruction tunes LLM on homogeneous graphs and studies the generalization to the graph size, graph description languages, node ID representation, and out-of-domain tasks. In contrast, we instruction tunes LLMs on heterogeneous graphs and design fine-grained sub-tasks to study the generalization. Besides, our work discusses the effect of graph representation on instruction tuning, which is not well-studied yet.

## 3 Instruction Tuning on Graph

### 3.1 Preliminaries

Formally, a general graph can be represented as $G = (V, E, T_V, T_E, \phi_V, \phi_E)$, where $V$ is the set of nodes, $E \subseteq V \times V$ is the set of edges, $T_V$ and $T_E$ are the sets of node types and edge types, and $\phi_V : V \to T_V$ and $\phi_E : E \to T_E$ are functions that map each node and edge to its respective type. To facilitate the expression of graph relationships, we introduce the notation $N(v)$ to denote the set of neighbors of node $v$, $P(u, v)$ to represent the set of paths connecting nodes $u$ and $v$, $p_{u,v}$ as a specific path between nodes $u$ and $v$, and $d(u, v)$ as the minimum number of edges on any path between nodes $u$ and $v$.

## 3.2 Task Definition

The core of instruction tuning is to involve as diverse a range of tasks as possible to enhance the model's generalization capabilities across different tasks. Therefore, in the context of graphs, we collect various graph tasks with diverse challenges, spanning from structural analysis to predictive inference. To comprehensively assess LLM's capabilities in addressing graph tasks, we categorize tasks according to their target answer type. The answer type delineates the nature of the output required for a graph task. We identify seven distinct answer types: **node**, **pair**, **count**, **boolean**, **path**, **graph** and **link prediction**. **Node** task seeks to identify specific nodes within the graph. **Pair** task seeks to identify node pairs connected by specific relationships or properties. **Count** task requires counting the number of certain nodes or paths. **Boolean** task provides a true/false answer to indicate the existence of specific structures. **Path** task necessitates finding a sequence of nodes that connect two specified nodes. **Graph** task demands extracting a subgraph represented as a set of node pairs. **Link prediction** task, different from previous answer types, aims to infer missing edges between nodes based on observed patterns of existing data.

For each answer type, we design a set of tasks, where each task requires a specific graph algorithm. All the tasks are listed in Table 1, along with their category, mathematical description, and an example. Furthermore, we subdivided each task into 2 to 4 **sub-tasks**, which share the same graph algorithm but focus on different node or edge types. For example, the *Find neighbors* task can be subdivided into sub-tasks like finding the brand of a product and finding the "also_view" product of a product. These fine-grained sub-tasks could facilitate a more detailed analysis of generalization, which will be introduced in Section 3.3.

| Answer Type | Task | Description | Examples |
|---|---|---|---|
| Node | Find neighbors | $\{v \in N(u)|\phi_E(u,v) \in T_E'\}$ | What are the products of brand1? |
| | Nodes shared neighbors | $\{v|\forall t_e \in T_E', \exists w \in V, \phi_E(u,w) = \phi_E(v,w) = t_e\}$ | What are the products that share "also_view" products with product1? |
| | N-hop neighbors | $\{v|\phi_V(v) \in T_V', d(u,v) <= c\}$ | What are the product nodes within 3-hop to product11? |
| Pair | Find pairs | $\{(v_1,v_2)|\phi(v_1,v_2) \in T_E', u \in \{v_1,v_2\}\}$ | What are the pairs connected by "also_view" edge and containing product11? |
| | Pairs shared neighbors | $\{(v_1,v_2)|\exists W \subset V : \forall w \in W, \phi_E(v_1,w) = \phi_E(v_2,w) \in T_E' \wedge |W| = c\}$ | What are the pairs that share 3 "also_buy" nodes? |
| Count | Degree count | $\{|V'||V' \subseteq N(u) : \forall v \in V', \phi_E(u,v) \in T_E'\}$ | How many "also_view" nodes does product11 have? |
| | Node count within N-hop | $\{|V'||V' \subset V : \forall v \in V', d(u,v) <= c\}$ | How many brand nodes are within 3-hop to product11? |
| | Path count | $\{|P'|P' \subseteq P(u,v)||\forall p_{u,v} \in P', len(p_{u,v}) = c\}$ | How many 3-hop simple paths exist between product11 and product12? |
| Bool | Linked by edge | $\{\phi_E(u,v) \in T_E'\}$ | Does a "also_view" edge exist between product11 and product12? |
| | Has path | $\{P(u,v) \neq \varnothing\}$ | Does a path exist between product11 and product12? |
| Path | Find paths | $\{P' \subseteq P(u,v)|len(p_{u,v}) = c, p_{u,v} \in P'\}$ | What are the 3-hop paths between product11 and product12? |
| | Shortest path | $\{p_{u,v}' \in P(u,v)|len(p_{u,v}') = \min(len(p_{u,v})|p_{u,v} \in P(u,v))\}$ | What are the shortest paths between product11 and product12? |
| Graph | Ego graph | $\{(v_1,v_2) \in E|d(u,v_1) <= c, d(u,v_2) <= c\}$ | What is the ego graph with radius 2 centered at product11? |
| Link Prediction | Link Prediction | $(E', T_E') \rightarrow \{0,1\}$ | Predict whether there is a "also_view" edge between product11 and product12. |

Table 1: The overview of all tasks. $T_E' \subseteq T_E$, $T_V' \subseteq T_V$ and $c \in \mathbb{Z}^+$ are the edge types, node types, and number restrictions in the task.

### 3.3 Evaluation Splits

To assess the generalization capabilities of the fine-tuned LLM on graph tasks, we propose three distinct types of unseen tasks: *unseen sub-tasks*, *unseen domain*, and *unseen answer type*. Each unseen type offers unique insights into the LLM's ability to adapt and perform on novel challenges beyond its training data.

**Unseen sub-tasks** evaluate the LLM's capacity to apply similar graph algorithms to sub-tasks slightly different from the ones seen during training. For instance, a model may be trained to find the shortest path between products and tested to find the shortest path between brands in an e-commerce network.

**Unseen domain** tasks evaluate the LLM's adaptability with graphs from out-of-domain networks. While the algorithms remain consistent with those learned during training, new node and edge types, and graph structures are introduced, testing the LLM's generalization across different domains.

**Unseen answer type** tasks push the boundaries of the LLM's capabilities by requiring it to generate answer types not encountered during training. Evaluating the model on these tasks assesses its capacity to innovate and extrapolate beyond its training data to develop new graph algorithms.

Generally, these three evaluation types collectively provide a comprehensive assessment of the LLM's generalization abilities across various dimensions of unseen tasks, which may bring useful insights into graph instruction tuning.

### 3.4 Data Collection

In this section, we outline the strategies and pipelines used to collect our dataset.

**Graph Sampling.** Given the impractical size of the original network against LLMs' limited context, we sample subgraphs from the original network and task LLMs over the subgraphs. To generate a subgraph, we sample an ego graph with a radius of 2, centered around a designated set of nodes. However, it's imperative to note that the number of nodes grows exponentially with each increment in hop count. Thus, we implement edge downsampling at each step. This downsampling process involves imposing a maximum limit on the number of edges for each type or establishing a ratio for downsampling. Different downsampling strategies can yield different graph structure distributions, which is useful for cross-domain generalization analysis.

**Node De-identification.** Given our objective of assessing LLMs' capacity for reasoning graph structures, textual information such as node names or titles becomes extraneous. To mitigate the potential influence of such textual data, we opt to de-identify nodes by representing them solely with their node type and a unique ID. For instance, a product in an e-commerce network might be denoted as "product11".

**Question-Graph Collection.** Each sample in our dataset contains a question as input and a graph as the context. Given that link prediction necessitates inductive reasoning, while the other answer types involve structure-based queries, distinct pipelines are developed to generate question-graph pairs.

For link prediction, we initiate by randomly sampling positive and negative samples in the form of (*head*, *relation*, *tail*) triples. Then, to augment the local structural understanding of the *head* and *tail* nodes, we sample a subgraph centered at each of these nodes.

Conversely, for the structure-based query tasks, we start by selecting two random nodes from the original graph and subsequently sampling subgraphs. Task-specific requirements dictate the identification of nodes within the subgraph that may harbor an answer, and graph algorithms are applied accordingly to uncover these answers. It is noteworthy that since the subgraph is sampled without considering the specific task, the resultant graph structure

remains independent of the task. Consequently, this approach facilitates the unbiased learning of general graph algorithms, irrespective of the graph's structural characteristics.

## 3.5 Graph Representation

Choosing the appropriate format for prompts is essential when utilizing LLMs, as it significantly affects the model's capacity to accurately interpret and process the information. We explore three primary prompt types: natural language, JSON, and DOT format.

Natural language prompts are versatile and intuitive for LLMs, offering a broad range of applications due to their human-like conversational style. Meanwhile, the JSON format for adjacency lists offers a structured, efficient means of information representation, aligning with LLMs' systematic processing capabilities for precise tasks. Additionally, the DOT format, a standard graph description language (code), enables a visual depiction of network relationships, beneficial for analyzing complex connections. We will delve deeper into their implications for LLM performance in Section 4.

## 3.6 Graph Instruction Tuning

To graph instruction tune the LLMs, we concatenate each sample's question $X_q$ with its graph representation $X_g$ to form the prompt and train the LLM to predict the answer $Y$ based on the prompt. We follow the implementation of Wang et al. (2023) to use the original auto-regressive training objective and mask the prompt tokens from loss computation. The loss function is

$$L = -\sum_i \log p_\theta(y_i | X_q, X_g, Y_{<i})$$

where $y_i$ is the $i$th token in the $Y$.

# 4  Experiments

|  | # Node | # Edge | Training size | Testing size | # Sub-Task | # Avg Nodes | # Avg Edges |
|---|---|---|---|---|---|---|---|
| Amazon MetaData | 2.07 m | 16.3 m | 22.4 k | 9.6 k | 40 | 87.14 | 111.63 |
| MAPLE | 2.15 m | 13.3 m | 21.84 k | 9.36 k | 39 | 67.09 | 74.18 |

Table 2: Dataset statistics

## 4.1  Experiment Settings

### 4.1.1  Datasets

We construct two separate domain graphs from two distinct datasets:

**Amazon Metadata**   (Ni et al., 2019) contains product metadata across 29 general categories on Amazon. We extract products, brands, and categories as nodes, connecting them via attributes "also_buy", "also_view", "brand", and "category". The graphs are built using metadata from the *CDs_and_Vinyl*, *Movies_and_TV*, and *Arts_Crafts_and_Sewing* categories.

**MAPLE**   (Zhang et al., 2023b) is derived from the Microsoft Academic Graph, featuring 19 scientific fields. We extract the authors, papers, and venues as nodes, and created edges using the "citation", "authorship", and "publication" relationships. This graph utilizes subjects *Political_Science*, *Computer_Science*, and *Geology* from this dataset.

We collect 800 samples for each sub-task and divide them into training and test sets with a ratio of 7:1. The statistics of the collected graphs and datasets are presented in Table 2.

#### 4.1.2 Models and Training

**Models** We perform graph instruction tuning with the Llama-2 7B (Touvron et al., 2023), Mistral 7B (Jiang et al., 2023), and Gemma 7B (Team et al., 2024) models and compared them with their instruction-tuned versions, which are not explicitly tailored to process structural information, to illustrate the benefits of our special graph instruction tuning.

**Fine-tuning** We employ LoRA (Hu et al., 2021) as our parameter-efficient fine-tuning approach. To ensure all models can access complete graph information, we train and test all models using samples that could fit within Llama-2's 4k context window. To assess the LLM's generalization to unseen sub-tasks and unseen domains, we train the models on part of the sub-tasks for each task, leaving the rest as the unseen sub-tasks. We also train the models on each domain separately, leaving the other domain as the unseen domain. We conduct a separate training for the evaluation of unseen answer types.

#### 4.1.3 Metrics

In our experiments, we evaluate performance using two key metrics, the Exact Match (EM) and the F1 score. Specifically, we use EM for the Count, Boolean, and Link prediction tasks, and F1 for the Node, Pair, Path, and Graph tasks. For the Path task, we treat each path as a single value and calculate the F1 score between the extracted and the ground truth paths.

### 4.2 Results

|       | Amazon | | | Maple | | |
| --- | --- | --- | --- | --- | --- | --- |
|       | # Avg Tokens | # Max Nodes | # Max Edges | # Avg Tokens | # Max Nodes | # Max Edges |
| NL   | 1869.56 | 226 | 324 | 1033.61 | 280 | 326 |
| JSON | 1972.44 | 199 | 289 | 1161.03 | 277 | 321 |
| DOT  | 2011.01 | 192 | 288 | 1181.22 | 277 | 321 |

Table 3: Statistics of different graph representations in 4k context

| | Amazon | | | | | | | | Maple | | | | | | |
| --- | --- | --- | --- | --- | --- | --- | --- | --- | --- | --- | --- | --- | --- | --- | --- |
| | Node | Pair | Count | Bool | Path | Graph | LP | AVG | Node | Pair | Count | Bool | Path | Graph | LP | AVG |
| | | | | | | | Baselines | | | | | | | | | |
| Llama-2-chat$_{NL}$ | 1.97 | **2.08** | 0.00 | 62.83 | 15.39 | 10.98 | 41.79 | 12.91 | 2.85 | **3.26** | **0.16** | 58.91 | 16.21 | 18.96 | 47.33 | 14.56 |
| Llama-2-chat$_{JSON}$ | 2.16 | 2.04 | 0.00 | 62.83 | 3.87 | 7.78 | 41.79 | 11.48 | 4.12 | 1.90 | 0.00 | 58.91 | 8.32 | 10.68 | 47.33 | 12.98 |
| Llama-2-chat$_{DOT}$ | **2.48** | 1.66 | 0.00 | 62.83 | 1.56 | 12.78 | 41.79 | 11.78 | 2.31 | 2.34 | 0.00 | 58.91 | 4.42 | **22.16** | 47.33 | 13.36 |
| Mistral-Inst$_{NL}$ | 0.01 | 3.98 | **12.89** | **37.55** | 18.15 | 7.27 | 58.21 | 13.84 | 0.03 | 5.81 | 14.04 | **42.98** | 20.54 | 6.70 | 52.67 | 15.04 |
| Mistral-Inst$_{JSON}$ | **2.91** | **8.38** | 12.43 | 37.30 | 12.29 | 8.86 | 58.21 | **14.75** | **4.24** | **10.81** | **14.45** | 41.09 | 9.74 | 7.74 | 52.67 | **15.77** |
| Mistral-Inst$_{DOT}$ | 1.65 | 5.28 | 8.00 | 37.17 | **18.43** | 12.50 | 58.21 | 14.02 | 3.07 | 4.85 | 12.41 | 41.09 | **23.34** | 9.72 | 52.67 | 15.74 |
| Gemma-Inst$_{NL}$ | 13.85 | 25.19 | 2.59 | **65.92** | **34.64** | 28.85 | 44.96 | 24.42 | 15.17 | 29.95 | 3.53 | **72.75** | **34.65** | 23.03 | **48.72** | 26.35 |
| Gemma-Inst$_{JSON}$ | 15.50 | 26.54 | **6.67** | 65.14 | 30.00 | 29.45 | 36.00 | 24.74 | 8.51 | 26.45 | **8.42** | 65.61 | 29.32 | 22.19 | 45.59 | 23.50 |
| Gemma-Inst$_{DOT}$ | **16.83** | **35.91** | 4.06 | 64.76 | 31.19 | **37.69** | **60.33** | **28.72** | **18.17** | **34.76** | 2.38 | 64.99 | 29.22 | **36.98** | 42.86 | **27.25** |
| | | | | | | | Finetuned | | | | | | | | | |
| Llama-2-GraphInst$_{NL}$ | 74.34 | 65.97 | 45.76 | 93.57 | **55.16** | 64.83 | 85.47 | 67.26 | 73.28 | 67.78 | 44.05 | 96.13 | 53.65 | 77.69 | 67.07 | 66.74 |
| Llama-2-GraphInst$_{JSON}$ | **80.20** | **68.49** | **46.48** | **96.48** | 52.75 | **65.39** | **85.02** | **69.46** | **75.33** | **68.42** | 46.62 | **98.11** | **55.21** | **80.06** | 64.42 | **68.30** |
| Llama-2-GraphInst$_{DOT}$ | 73.64 | 64.26 | 43.76 | 91.20 | 50.07 | 61.39 | 77.11 | 64.69 | 70.59 | 63.69 | **47.32** | 94.35 | 54.47 | 74.83 | **69.08** | 65.86 |
| Mistral-GraphInst$_{NL}$ | 87.43 | 75.38 | 48.47 | 98.13 | 66.55 | 80.09 | 75.23 | 75.16 | **86.17** | 76.39 | 48.86 | 97.80 | 68.55 | **86.14** | 76.97 | **75.69** |
| Mistral-GraphInst$_{JSON}$ | **89.63** | **81.18** | **50.77** | **98.73** | 62.16 | **83.32** | 76.15 | **77.11** | 82.96 | **79.58** | 50.94 | **99.16** | 68.61 | 84.13 | 75.95 | 75.64 |
| Mistral-GraphInst$_{DOT}$ | 86.30 | 72.91 | 46.24 | 96.97 | **68.71** | 81.01 | **77.98** | 74.44 | 79.01 | 74.96 | 50.74 | 98.95 | 66.47 | 85.69 | **79.02** | 74.03 |
| Gemma-GraphInst$_{NL}$ | 87.34 | 76.30 | 46.51 | 97.36 | **68.06** | 77.31 | **85.17** | 75.43 | **88.90** | 74.76 | 47.84 | 97.49 | 61.50 | 86.04 | **77.53** | 75.20 |
| Gemma-GraphInst$_{JSON}$ | **90.15** | **78.11** | 49.98 | **99.23** | 65.68 | 78.08 | 82.42 | **76.98** | 88.50 | **75.33** | **51.74** | **98.64** | 63.39 | 83.15 | 70.91 | **75.50** |
| Gemma-GraphInst$_{DOT}$ | 87.43 | 78.09 | 47.96 | 96.36 | 67.43 | **83.50** | 83.94 | 76.37 | 85.75 | 74.00 | 50.14 | 98.43 | **68.14** | **88.71** | 70.77 | 75.29 |

Table 4: Experimental results of three graph representations, including natural languages (NL), JSON, and DOT.

#### 4.2.1 Graph Representation

**Scalability** Table 3 presents the average length in tokens and the maximum graph size in a 4k context concerning node and edge number for each of the three graph representations, natural language (NL), JSON, and DOT, in the two datasets. It is shown that natural language has the most compact representation and can handle the largest graph in a limited context budget.

**Performance** Table 4 presents the performance of both vanilla LLMs and graph instruction-tuned LLMs on the test sets of two datasets. The results reveal notable improvements in all tasks when comparing the graph instruction-tuned models with their text instruction-tuned counterparts. This suggests that the LLMs fine-tuned on our benchmark exhibit an enhanced understanding of graph structures, leading to improved reasoning capabilities for answering questions. Notably, the graph representations in JSON format consistently outperform those in other formats across various tasks, yielding the best overall performance for all three models.

Furthermore, we conduct studies to compare how different graph representations perform with different scales of LLMs. Concretely, we instruction-tune both Llama-2-7b and Llama-2-13b on the Amazon dataset with the three graph representations. As illustrated in Figure 2, the observation is in line with our previous finding that JSON format is the best bridge for LLMs interacting with graphs and can yield the best performance on both scales.

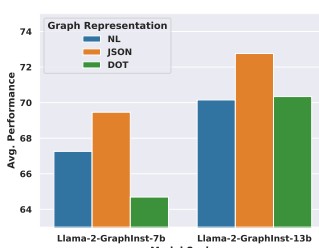

Figure 2: Compare LLMs of different scales using three graph representations.

We postulate that this superiority of JSON representations stems from their clearer structural depiction compared to natural language. Moreover, JSON is a more prevalent format in the pre-training data compared to DOT. Consequently, models trained with graph instruct-tuning tend to find JSON particularly effective in comprehending and reasoning about complex graph structures.

Given the consistently strong performance associated with the use of JSON format, our analysis in the following sections focuses primarily on models trained with JSON format.

#### 4.2.2 Sub-task Generalization

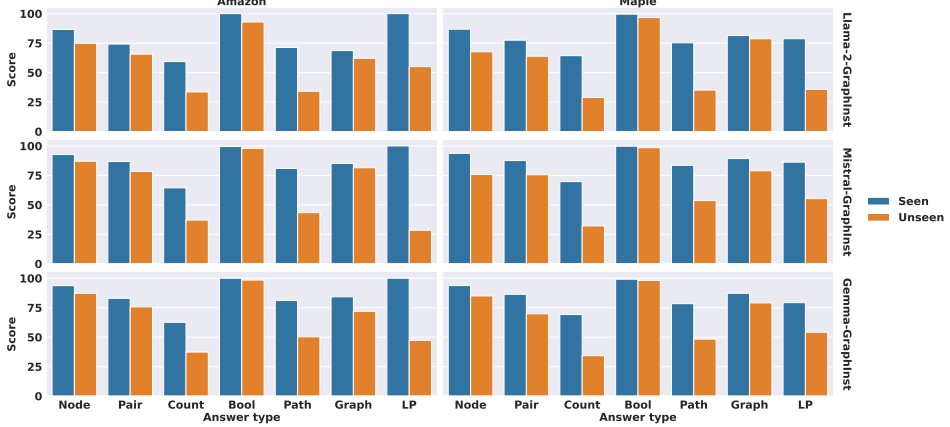

Figure 3: Experiment results of sub-task generalization on two datasets.

As mentioned in Section 3.3, we show the performance of models over the in-domain seen and unseen sub-tasks under each answer type. In Figure 3, all models present an excellent generalization on the unseen sub-tasks of Node, Pair, Bool, and Graph tasks, with a small

drop in the performance, but fail to generalize to the unseen sub-tasks of Count, Path, and Link prediction tasks. We examine the failure cases and conclude the following reasons for the failure of these three types.

For the Count task, the greatest performance drops occur in sub-tasks of *Degree count*, which require only single-hop information of the queried node. Compared to the sub-tasks of *Find neighbors*, which also requires only single-hop information but returns a set of nodes, the Count answer type is more abstract, and thus, overfits the model to the seen sub-tasks. For the Path task, the most significant cause of failure in unseen sub-tasks is the incorrect starting nodes in the generated paths. For example, for the Llama-2 model trained on the MAPLE dataset, around 77.12% of the generated paths fail to start with the queried source node and 71.46% of the failure cases start with the paper node, which is the source node type in the seen sub-tasks. Our manual checking confirms that the model can still generate partially correct paths in the unseen sub-tasks, indicating that the model does learn the path-finding algorithm, but fails due to the overfit in the starting node. For the link prediction task, this failure makes sense because the model is only trained to infer the existence of a subset of edge types, while different edge types may be inferred from different graph patterns, the model fails to generalize this inductive reasoning to the unseen edge types.

### 4.2.3 Domain Generalization

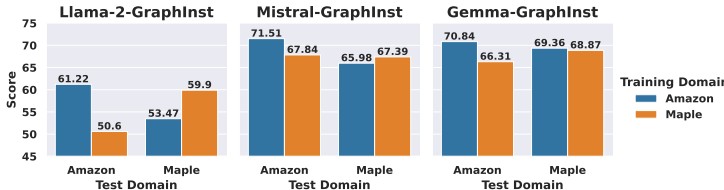

Figure 4: Compare LLMs of different scales on domain generalization.

To evaluate the domain generalization of the instruction-tuned model, we compare the models separately trained on the two datasets by their performance on the unseen sub-tasks of both in-domain and out-of-domain datasets. This approach allows us to assess their performance on unseen tasks in cross-domain scenarios. Figure 4 demonstrates the averaged performance of all unseen sub-tasks in the corresponding dataset except for the link prediction due to the conclusion from Section 4.2.2. In most scenarios, the model trained on a different domain has an acceptable performance drop compared to the model trained on the tested domain. In addition, the models trained on the Amazon Metadata network have a smaller performance drop (6.43% for Llama-2, 1.41% for Mistral and -0.49% for Gemma) than the models trained on the Maple network (10.62% for Llama-2, 3.67% for Mistral and 4.53% for Gemma). According to the statistics in Table 2, the subgraphs from the Amazon Metadata network are generally larger than the subgraphs from the Maple network. This may indicate that training on larger graphs can better generalize the model to smaller out-of-domain graphs.

### 4.2.4 Answer type Generalization

In Table 5, we aim to assess the capacity of instruction-tuned LLMs for generalizing across different answer types. As highlighted in Section 3.3, this represents a particularly challenging scenario due to the potentially large discrepancy between the training tasks and the testing tasks. To this end, we specifically exclude *Pair*, *Bool* and *Graph* from the training dataset, and subsequently instruction-tune the LLM as *Mistral-GraphInst-masked*.

Regarding the results, *Mistral-GraphInst-masked* indicates compromised performance on the unseen *Pair*, *Bool*, and *Graph* tasks, a direct consequence of their absence during training. Despite this, it still manages to surpass *Mistral-Inst*, which is not fine-tuned with graph structures, in terms of performance. The findings suggest that our instruction design

effectively enables the LLM to grasp structural information and apply it to successfully tackle questions beyond its initial training scope.

| | Amazon | | | | | |
|---|---|---|---|---|---|---|
| | **Node** | **Pair**[*] | **Count** | **Bool**[*] | **Path** | **Graph**[*] |
| Mistral-Inst$_{JSON}$ | 2.91 | 8.38 | 12.43 | 37.30 | 12.29 | 8.86 |
| Mistral-GraphInst$_{JSON}$ | 89.63 | 81.18 | 50.77 | 98.73 | 62.16 | 83.32 |
| Mistral-GraphInst-masked$_{JSON}$ | 88.09 | 56.43 | 49.91 | 90.18 | 59.31 | 53.65 |
| | Maple | | | | | |
| Mistral-Inst$_{JSON}$ | 4.24 | 10.81 | 14.45 | 41.09 | 9.74 | 7.74 |
| Mistral-GraphInst$_{JSON}$ | 82.96 | 79.58 | 50.94 | 99.16 | 68.61 | 84.13 |
| Mistral-GraphInst-masked$_{JSON}$ | 79.70 | 40.90 | 50.48 | 77.15 | 64.30 | 36.64 |

Table 5: Answer Type Generalization, where tasks *Pair*, *Bool* and *Graph* are unseen when training *Mistral-GraphInst-masked*.

#### 4.2.5 Case Study

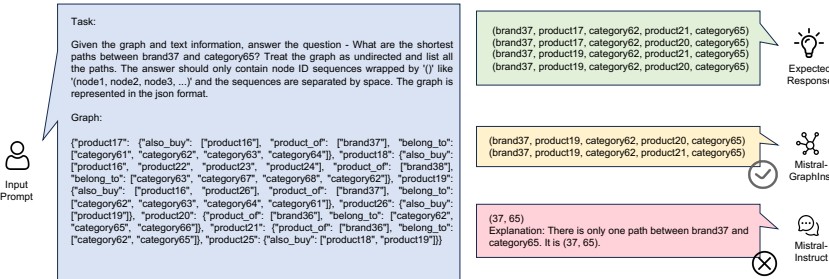

Figure 5: Case study on finding the shortest path between two non-product nodes in the Amazon dataset, depicted through a graph in JSON format.

To demonstrate the effectiveness of graph-based instruction tuning, we explore the model's effectiveness in identifying the shortest path between two non-product nodes within an undirected graph from the Amazon dataset. This task highlights the significant challenges faced by LLMs, including the need to comprehend complex graph structures and apply graph theory algorithms within a computational environment that requires processing large amounts of data and evaluating many pathways at once.

As depicted in Figure 5, *Mistral-GraphInst*, with its specialized tuning for graph dataset analysis, can better overcome these challenges by bridging the gap between natural language and computational graph theory. Unlike *Mistral-Instruct* mainly optimized for broad language tasks, *Mistral-GraphInst* is adept at navigating the intricacies of graph structures, enabling it to perform sophisticated analyses like shortest path discovery with higher precision and efficiency. Despite still occasionally missing a few shortest paths, *Mistral-GraphInst*'s capability of handling complex network dynamics positions it as a superior tool for tasks demanding in-depth exploration of graphs, thereby advancing our ability to interpret and analyze complex data structures.

## 5 Conclusion

In this paper, we investigate instruction-tuning LLMs on graph-related tasks. We first construct a dataset that contains comprehensive graph-related tasks from the academic and e-commerce domains. We then conduct extensive experiments to explore the best representation for LLMs to understand graphs and gain insights into the generalization of graph instruction-tuned LLMs to different kinds of unseen tasks. Future studies can consider applying such instruction-tuning techniques to graphs from other domains.

**Acknowledgments**

The research was supported in part by US DARPA KAIROS Program No. FA8750-19-2-1004 and INCAS Program No. HR001121C0165, National Science Foundation IIS-19-56151, and the Molecule Maker Lab Institute: An AI Research Institutes program supported by NSF under Award No. 2019897, and the Institute for Geospatial Understanding through an Integrative Discovery Environment (I-GUIDE) by NSF under Award No. 2118329. Any opinions, findings, and conclusions or recommendations expressed herein are those of the authors and do not necessarily represent the views, either expressed or implied, of DARPA or the U.S. Government.

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
