# OpenReview forum: "Investigating Instruction Tuning Large Language Models on Graphs"
_colmweb.org/COLM/2024/Conference — COLM_

### Official Review · Reviewer_zfbk · 2024-04-19

**Rating:** 6
**Confidence:** 4
**Ethics Flag:** 1

**Summary:**

This paper explores the application of LLMs to graph-related tasks through instruction tuning. A dataset including 14 graph tasks and approximately 63k question-graph pairs across academic and e-commerce domains is utilized to evaluate different graph representations and their impact on LLM performance. The study identifies JSON as the most effective format, achieving superior results across a range of graph types and tasks. It also examines the generalization capabilities of LLMs across three dimensions: unseen sub-tasks, domains, and answer types. The findings indicate that while LLMs exhibit strong generalization in some areas, they face challenges in others, notably inductive reasoning tasks such as link prediction.

**Questions To Authors:**

1. Given the wide variety of graph datasets available, what motivated the choice of Amazon Metadata and MAPLE for your evaluation? Could you elaborate on how these particular datasets align with the objectives of your study?
2. Regarding scalability, could you detail the maximum graph size that your graph representation methods can effectively handle? What are the limitations in terms of node and edge count for each representation method?
3. The performance figures for LLaMA-2 and Mistral models without instruction tuning appear unusually low in Table 3, and seems like random guess. Could you provide some insights?

**Reasons To Accept:**

1. The exploration of representing graphs in JSON and DOT formats is innovative and provides valuable insights into optimizing LLM performance across complex graph structures.
2. The examination of domain generalization offers meaningful contributions to the field, demonstrating how LLMs can adapt to different graph structures and domains without prior exposure.
3. This paper adeptly highlights both the potential and the limitations of applying LLMs to complex graph-based problem-solving.

**Reasons To Reject:**

1. The methodology employed in graph instruction tuning is inadequately detailed and only briefly introduced in section 4.1.2. To enhance clarity and understanding, this method should be thoroughly described earlier in the paper, ideally in section 3, alongside a comprehensive explanation of the techniques and processes involved in graph instruction tuning, e.g., which loss is used.
2. The paper lacks a broad comparative analysis. It only evaluates the performance of two LLM backbones, neglecting other popular language model variants, such as those from the GPT series, which could serve as valuable baselines. Additionally, comparing these results with traditional graph methods and graph neural networks would significantly enrich the study’s insights and implications for the field.
3. There is no analysis of the scalability of the methods, which is a critical aspect to consider for practical applications. It is unkown the maximum graph size the methods can handle.

---

> ### Author Rebuttal · Authors · 2024-05-31
>
> Reason to reject
> 1. Thank you for your suggestion. In our final version, we will update this part by adding more details about the graph instruction tuning.
> 2. Thank you for suggesting a broader comparative analysis. We have recently conducted some experiments with another LLM, Google’s Gemma-7b. The current results on the Amazon Metadata network also support our following observation in the paper:
> JSON format gives the LLMs the best performance after tuning.
> The model presents an excellent generalization on the unseen sub-tasks of Node, Pair, Bool, and Graph tasks, with a small drop in the performance, but fails to generalize to the unseen sub-tasks of Count, Path, and Link prediction tasks.
> More models and results will be added to the final version of the paper.
>
> Questions To Authors:
> 1. One of our research questions is “To what extent can LLMs fine-tuned on limited graph-related tasks generalize to unseen tasks?”, the graph-related tasks can be “limited” from several perspectives: limited queried information (only a few node types and edge types), limited graph structure distribution (a few distinct domains) and limited trained graph algorithm. Therefore, we would like to choose graph datasets that have only a few node and edge types and the datasets should be from distinct domains. Also, to study the LLM generalization capability with its general world knowledge, we would like the graph datasets to contain some general types, like paper and products, rather than some specialized types like diseases and pharmacology. Therefore, we choose the Amazon Metadata and MAPLE, where both contain 3 to 4 common node and edge types and have different graph structure distributions against each other.
>
> 2. We have collected some statistics of the graphs in both datasets. Below are the statistics of the Amazon Metadata training dataset using JSON format:
> max node #: 176,
> max edge #: 276,
> 80% have between 29 - 107 nodes,
> 80% have between 30 - 140 edges.
> We also observe that the instruction-tuned LLM’s performance on different tasks decreases differently as the number of nodes or edges increases. We will include more detailed data and discussion in our final version.
>
> 3. In most of the failure cases, the outputs are some meaningless text like “node1, node2, …”. We hypothesize that this is because LLM didn't understand the graph structure from our prompt. We will try different prompts with more detailed explanations and report any influence in the final version.

---

> > ### Author Response · Authors · 2024-06-01
> > **Some current experiment results of Gemma-7b on Amazon Metadata dataset**
> >
> > **Add to Table 3:**
> >
> > |                         | Node  | Pair  | Count | Bool  | Path  | Graph | LP    | AVG   |
> > | ----------------------- | ----- | ----- | ----- | ----- | ----- | ----- | ----- | ----- |
> > | Gemma-Inst_natural | 13.85 | 25.20 | 2.59  | **65.92** | **34.64** | 28.85 | 44.96 | 24.42 |
> > | Gemma-Inst_json    | 15.50 | 26.54 | **6.67**  | 65.15 | 30.00 | 29.45 | 36.00 | 24.74 |
> > | Gemma-Inst_dot     | **16.83** | **35.91** | 4.06  | 64.76 | 31.19 | **37.69** | **60.33** | **28.72** |
> > | Gemma-GraphInst_natural | 87.34 | 76.30 | 46.51 | 97.36 | **68.06** | 77.32 | **85.17** | 75.43 |
> > | Gemma-GraphInst_json    | **90.15** | **78.11** | **49.98** | **99.24** | 65.68 | 78.08 | 82.42 | **76.98** |
> > | Gemma-GraphInst_dot     | 87.43 | 78.09 | 47.96 | 96.36 | 67.43 | **83.51** | 83.94 | 76.37 |
> >
> > The results show JSON format provides the best performance after graph instruction tuning.
> >
> > **Add to Figure 3:**
> > |                      |        | Node  | Pair  | Count | Bool   | Path  | Graph | LP     | AVG   |
> > | -------------------- | ------ | ----- | ----- | ----- | ------ | ----- | ----- | ------ | ----- |
> > | Gemma-GraphInst_json | Seen   | 93.69 | 82.97 | 62.60 | 100.00 | 81.07 | 84.22 | 100.00 | 85.02 |
> > |                      | Unseen | 87.19 | 75.69 | 37.36 | 98.47  | 50.30 | 71.94 | 47.25  | 69.71 |
> >
> > The results show the graph instruction tuned LLM generalizes well on unseen sub-tasks of Node, Pair, Bool, and Graph tasks, with a small drop in the performance, but fails to generalize to the unseen sub-tasks of Count, Path, and Link prediction tasks.

---

> ### Comment · Reviewer_zfbk · 2024-06-04
> **Response of Reviewer zfbk**
>
> > We have recently conducted some experiments with another LLM, Google’s Gemma-7b. The current results on the Amazon Metadata network also support our following observation in the paper: JSON format gives the LLMs the best performance after tuning. The model presents an excellent generalization on the unseen sub-tasks of Node, Pair, Bool, and Graph tasks, with a small drop in the performance, but fails to generalize to the unseen sub-tasks of Count, Path, and Link prediction tasks. More models and results will be added to the final version of the paper.
>
> You only test three small-scale LLMs less than 10B. From my opinion, which data format for graphs is better is more likely model-independent and a such conclusion may be misleading. You can also recommended to test larger LLMs or APIs to convince the readers.
>
> > In most of the failure cases, the outputs are some meaningless text like “node1, node2, …”. We hypothesize that this is because LLM didn't understand the graph structure from our prompt. We will try different prompts with more detailed explanations and report any influence in the final version.
>
> From my experience with LLMs for graph problems, this behavior seems unusual. It likes the behavior of LLaMA 2 7b without the chat version. The chat version can follow your insturctions and make a prediction. Please double check it, and also consider providing some examples in prompts.
>
> Overall, I think the paper has some merits and interesting findings that worth an accept. So I raised my score.

---

### Official Review · Reviewer_zHpC · 2024-05-11

**Rating:** 6
**Confidence:** 4
**Ethics Flag:** 1

**Summary:**

This paper presents a benchmark to evaluate the ability of large language models (LLM) in understanding graph structures. This benchmark contrains a diverse collection of 79 graph-related tasks from academic and e-commerce domains, featuring 44,240 training instances and 18,960 test samples. Besides the benchmark, the author also study the impact of different graph representations in affecting the performance of LLM and find that JSON is a better choice. In addition to that, the author also study how LLM can generalize to unseen examples.

**Questions To Authors:**

1. Can you compare more LLMs beyond Llama 2 and Mistral?

2. Can you elaborate on the technique contribution?

3. What is the gap between small models on graphs and LLM in understanding graph structure?

**Reasons To Accept:**

1. Present a solid benchmark containing a diverse collection of 79 graph-related tasks from academic and e-commerce domains, featuring 44,240 training instances and 18,960 test samples.

2. The paper is good written and easy to follow. Every detail is presented clearly and reading friendly for unfamiliar readers.

3. The analysis about different graph representations and the ability on unseen examples gives new insights.

**Reasons To Reject:**

1. The proposed benchmark only consider network data in academy and e-commerce and ignore widely used graph data in the domain of chemistry and biology.

2. About novelty. This is not a brand new benchmark. Many previous works has already considered the ability of LLM in understanding graph structures. For example

3. The paper only evaluates the ability of Llama2 and Mistral, which is hard to cover all representive LLMs.

4. Limited technique contribution. This paper only gives evaluation results and insights and the technique used here on how to integrate LLM and graph follows previous methods.

---

> ### Author Rebuttal · Authors · 2024-05-31
>
> Reasons To Reject:
> 1. Thank you for your suggestion! We agree that there are many networks from domains like chemistry and biology, which can make the benchmark more diverse and raise more general interests. Since one of our research questions is “To what extent can LLMs fine-tuned on limited graph-related tasks generalize to unseen tasks?”, the graph-related tasks can be “limited” for limited queried information (only a few node types and edge types) or limited graph structure distribution (a few distinct domains). This is why we choose the Amazon Metadata and MAPLE, where both contain 3 to 4 common node and edge types and have different graph structure distributions against each other. We will try to include more networks in our dataset if they satisfy these requirements.
>
> 2. We may have different ideas at this point. Our benchmark is built for two purposes: first, to find the optimal graph representation that helps LLMs learn graph structures effectively; second: to analyze the generalization capability of LLMs fine-tuned on limited graph-related tasks to unseen tasks. These are the novelty of our work.
>
> Questions To Authors:
> 1. Yes, we have recently conducted some experiments with another LLM, Google’s Gemma-7b. The current results on the Amazon Metadata network also support our following observation in the paper: JSON format gives the LLMs the best performance after tuning. The model presents an excellent generalization on the unseen sub-tasks of Node, Pair, Bool, and Graph tasks, with a small drop in the performance, but fails to generalize to the unseen sub-tasks of Count, Path, and Link prediction tasks. More models and results will be added to the final version of the paper.
>
> 2. Our technique contributions are:
> + We create a benchmark for investigating the influence of different graph representations in graph instruction tuning and the generalization capability of the LLM instruction tuned on limited graph-related tasks.
> + Our result shows the JSON format gives the LLMs the best performance after tuning.
> + Our experiments on generalization reveal tasks where LLM could be overfitted or hard to generalize. Our experiments also show LLM can derive algorithms from the learned algorithms.
>
> 3. We think the small models could be fine-tuned to each specific task with a specific network but is hard to generalize to a different task. On the other hand, LLM with strong capability in reasoning and understanding, can generalize better in unseen tasks.

---

> ### Author Response · Authors · 2024-06-01
> **Some current experiment results of Gemma-7b on Amazon Metadata dataset**
>
> **Add to Table 3:**
>
> |                         | Node  | Pair  | Count | Bool  | Path  | Graph | LP    | AVG   |
> | ----------------------- | ----- | ----- | ----- | ----- | ----- | ----- | ----- | ----- |
> | Gemma-Inst_natural | 13.85 | 25.20 | 2.59  | **65.92** | **34.64** | 28.85 | 44.96 | 24.42 |
> | Gemma-Inst_json    | 15.50 | 26.54 | **6.67**  | 65.15 | 30.00 | 29.45 | 36.00 | 24.74 |
> | Gemma-Inst_dot     | **16.83** | **35.91** | 4.06  | 64.76 | 31.19 | **37.69** | **60.33** | **28.72** |
> | Gemma-GraphInst_natural | 87.34 | 76.30 | 46.51 | 97.36 | **68.06** | 77.32 | **85.17** | 75.43 |
> | Gemma-GraphInst_json    | **90.15** | **78.11** | **49.98** | **99.24** | 65.68 | 78.08 | 82.42 | **76.98** |
> | Gemma-GraphInst_dot     | 87.43 | 78.09 | 47.96 | 96.36 | 67.43 | **83.51** | 83.94 | 76.37 |
>
> The results show JSON format provides the best performance after graph instruction tuning.
>
> **Add to Figure 3:**
> |                      |        | Node  | Pair  | Count | Bool   | Path  | Graph | LP     | AVG   |
> | -------------------- | ------ | ----- | ----- | ----- | ------ | ----- | ----- | ------ | ----- |
> | Gemma-GraphInst_json | Seen   | 93.69 | 82.97 | 62.60 | 100.00 | 81.07 | 84.22 | 100.00 | 85.02 |
> |                      | Unseen | 87.19 | 75.69 | 37.36 | 98.47  | 50.30 | 71.94 | 47.25  | 69.71 |
>
> The results show the graph instruction tuned LLM generalizes well on unseen sub-tasks of Node, Pair, Bool, and Graph tasks, with a small drop in the performance, but fails to generalize to the unseen sub-tasks of Count, Path, and Link prediction tasks.

---

> ### Comment · Reviewer_zHpC · 2024-06-05
>
> Thank you for your detailed response. Most of my concerns have been resolved. I will maintain the score.

---

### Official Review · Reviewer_172j · 2024-05-13

**Rating:** 6
**Confidence:** 3
**Ethics Flag:** 1

**Summary:**

Authors propose a new dataset of graphs with associated tasks to answer given a graph representation. Authors use LORA to adapt LLM to enhance its understanding of the graphs. Graphs represented by one of three formats.

Authors conclude that JSON is the best format with advnatage for adapted LLMs in performance.

**Questions To Authors:**

(1) Could you have written the queries in natural language, is there a need to use phrases such as 'also_buy' or 'product_of'?

(2) Was the LLM allowed to generate tokens that are out of the possible set of answers? For example, if the task is counting, could the LLM generate an answer than is not in digits? Could you have tested the performance of the LLM scoring the possible outcomes instead of generating novel responses?


(3)

**Reasons To Accept:**

(1) Understanding strcutured data within the context of LLMs is quite important with wide set of applications

(2) Authors propose a useful dataset that will help other researchers studying the problem.

**Reasons To Reject:**

(1) Graph representation approaches proposed are all rule based, it would have been great to propose a learnable approach to graph such as an encoder based representation.

(2) The datasets are small and diverse enough.

---

> ### Author Rebuttal · Authors · 2024-05-31
>
> Reasons To Reject:
> 1. Using a graph encoder to generate the graph representation is an interesting and novel approach. However, currently, there are many works still using the rule-based linearization approach, which can directly make use of the knowledge in the popular decoder-only LLM. Therefore, one of our goals in the paper is to provide insights into how different rule-based graph representations will affect the performance under the setting of graph instruction tuning so that it may help other researchers when choosing their graph representation.
> 2. Maybe this is not a drawback? Since LLM can be easily adapted to specific tasks with a few thousand training examples, a small and diverse dataset may be appropriate to study the generalization of LLM in graph-related problems.
>
> Questions To Authors:
> 1. Yes, we do have a few queries written in natural language. For example, when asking for the products of a brand, the query is formulated as “What are the products of brandxxx?”. We mostly use the edge types because we can use some templates to generate different sub-tasks that share the same graph algorithm but focus on different node types or edge types.
> 1. This is an interesting direction to explore! However, we didn’t allow this. We test the LLM’s understanding of graph and graph algorithm by evaluating how well it can generate correct answers for any graph-related query so that we can decide whether LLM can be trusted as an independent graph problem solver. Therefore, we didn’t consider the approach of scoring over the possible outcomes. Since this may involve changing the dataset, we may not be able to test this during the rebuttal, but we may consider this as a future work.

---

### Decision · Program_Chairs · 2024-07-10

**Decision:**

Accept

**Comment:**

The paper introduces a valuable dataset and benchmark for evaluating large language models (LLMs) on graph-related tasks, which is a significant contribution given the importance of understanding structured data in various applications. The dataset is diverse, covering 79 graph-related tasks from academic and e-commerce domains, and provides a comprehensive evaluation platform for future research. The study also innovatively explores different graph representation formats, concluding that JSON is the most effective for enhancing LLM performance. Additionally, the paper examines the generalization capabilities of LLMs across unseen sub-tasks, domains, and answer types, offering insightful analyses and highlighting both potential and limitations in applying LLMs to complex graph-based problem-solving. The clear and accessible writing style makes the paper easy to follow, even for readers unfamiliar with the topic.

Despite its contributions, the paper has several limitations. The proposed graph representation approaches are all rule-based, lacking a learnable approach such as encoder-based representations, which could have added more depth to the study. The methodology for graph instruction tuning is inadequately detailed, and the lack of a comprehensive comparative analysis with other popular LLM variants and traditional graph methods diminishes the robustness of the findings. The paper also overlooks scalability issues, failing to address the maximum graph sizes the methods can handle, which is crucial for practical applications. Furthermore, the benchmark focuses only on academic and e-commerce domains, neglecting other important areas like chemistry and biology. The limited evaluation of only two LLMs (LLaMA-2 and Mistral) restricts the generalizability of the results, and the paper's overall technical contributions are minimal, primarily offering evaluation results and insights rather than novel methodologies.

[comments from the PCs] It's advised to discuss scalability issues in the paper, as the AC notes this important aspect is overlooked.